# Preeclampsia and its determinants in Ethiopia: A systematic review and meta-analysis

**Bekalu Getnet Kassa** [1]*, **Sintayehu Asnkew**[2], **Alemu Degu Ayele**[1], **Azezu Asres Nigussie**[3], **Basaznew Chekol Demilew**[4], **Gedefaye Nibret Mihirete**[1]

1 Department of Midwifery, College of Health Sciences, Debre Tabor University, Debre Tabor, Ethiopia,
2 Department of Psychiatry, College of Health Sciences, Debre Tabor University, Debre Tabor, Ethiopia,
3 Department of Midwifery, College Medicine and Health Sciences, Bahir Dar University, Bahir Dar, Ethiopia,
4 Department of Anesthesia, College of Health Sciences, Debre Tabor University, Debre Tabor, Ethiopia

* bekalugetnet947@gmail.com

**Data Availability Statement:** All relevant data are within the paper and its Supporting Information files.

## Abstract

### Background

Preeclampsia is a serious condition that is linked to poor perinatal outcomes. In Ethiopia, the overall prevalence of preeclampsia and its associated factors is uncertain. Therefore, the purpose of this review was to find the prevalence of pre-eclampsia and its determinants in Ethiopia.

### Methods

To find primary studies, PubMed, Google Scholar, HINAR, Scopus, the Web of Sciences, and grey literature searches were used between January 1, 2013, and January 1, 2023, in Ethiopia. A Microsoft Excel sheet was used to extract data. The pooled prevalence of pre-eclampsia was predicted using a random-effect model.

### Results

Twenty-nine studies were included. The pooled prevalence of pre-eclampsia was 11.51% (95% CI: 8.41, 14.61). Age > 35 years old (AOR = 2.34, 95%CI, 1.74–2.94; p-value = 0.64), housewife (AOR = 2.76, 95%CI, 1.2–4.32; p-value = 0.37), previous history of pre-eclampsia (AOR = 4.02, 95%CI, 2.91–5.55; p-value = 0.09), family history of hypertension (OR = 1.84, 95%CI, 1.39–2.3; p-value = 0.4), history of chronic hypertension (AOR = 2.44, 95%CI, 1.8–3.08; p-value = 0.67), history of multiple pregnancies (AOR = 1.45, 95%CI, 1.09–1.8; p-value = 0.38), and alcohol intake during pregnancy (AOR = 1.53, 95%CI, 1.03–2.04; p-value = 0.03) were the determinants of pre-eclampsia.

### Conclusions

When compared to previous studies, the overall pooled prevalence of pre-eclampsia was high. Pre-eclampsia is associated with maternal age >35 years, being a housewife, having a history of preeclampsia, having a history of chronic hypertension, having a family history of

**Funding:** The author(s) received no specific funding for this work.

**Competing interests:** The authors have declared that no competing interests exist.

**Abbreviations:** AOR, Adjusted Odd Ratio; BMI, : Body Mass Index; DM, Diabetes Mellitus; GDM, Gestational Diabetes Mellitus; SNNPR, South Nation Nationality People Region; WHO, World Health Organization.

hypertension, having diabetes mellitus, drinking alcohol during pregnancy, and having multiple pregnancies.

## Introduction

Pre-eclampsia is the most prevalent and severe kind of pregnancy-induced hypertension and can affect every organ system [1, 2]. Diabetes, renal disease, obesity, repeated pregnancies, primiparity, age beyond 35 and younger than 20 years, urinary tract infection, personal or family history of preeclampsia, chronic hypertension, renal disease, and autoimmune disorder, the prolonged interval between pregnancies and the history of abortion all are risk factors for pre-eclampsia [3]. Alcohol intake during pregnancy, body mass index $\geq$30 kg/m$^2$, unwanted pregnancy [4], psychological distress [5], traditional medicine utilization during pregnancy [6], low birth weight, low APGAR score, preterm birth, and fetal death [7] were associated with pregnancy-induced hypertension.

Global studies reveal that the incidence of pre-eclampsia in nulliparous women ranged from 3 to 10% while in multiparous women, the incidence of preeclampsia ranged from 2 to 5% [8–10]. The incidence of pre-eclampsia is also influenced by race, ethnicity, and genetic predisposition. In accordance with the studies, the incidence of pre-eclampsia was 5% in white women, 9% in Hispanic women, and 11% in African Americans [11]. But antenatal advice about nutrition, vegetable intake and fruit intake during pregnancy were protective factors for pre-eclampsia results reduce the incidence [12, 13].

Globally, pregnancy-induced hypertension is the second leading cause of maternal mortality, contributing to around 14% of maternal deaths globally [14]. It is responsible for 16% of maternal mortality in Sub-Saharan Africa and 16.9% of maternal mortality in Ethiopia. Preeclampsia and eclampsia are associated with hypertension and are known to poorly impact maternal and newborn mortality and morbidity [15].

Pre-eclampsia causes several adverse pregnancy consequences. It increases the number of caesarian deliveries, hospital stays, postpartum hemorrhage, hemolysis, elevated liver function tests, low platelet count, placental abruption, and heart failure in the mother [12, 15]. Most pre-eclampsia and eclampsia deaths are prevented by providing early identification, diagnosis, and efficient treatment to women. Improving the ability of the healthcare system to prevent and treat hypertension in women is an essential step toward reducing maternal and newborn morbidity and mortality.

In developing countries, the prevalence of pre-eclampsia ranges from 1.8 to 16.7% [16, 17]. In Ethiopia, it as well varies from 1.2% to 19.1% [18]. Ethiopia's government has taken multiple measures to reduce maternal and newborn morbidity and death by increasing access to and improving facility-based maternity and newborn services. However, maternal morbidity and mortality related to pregnancy-induced hypertension remains to be a global agenda [19].

In Ethiopia, several primary studies have been conducted in different areas to examine this problem. These studies had great discrepancies and inconsistent results across various regions in the state. Hence, this systematic review and meta-analysis aimed to synthesize the pooled prevalence of pre-eclampsia and its associated factors in Ethiopia. Thus, this study will provide input information for policymakers, and concerned shareholders for designing prevention strategies, and management of pre-eclampsia and its complication. Furthermore, the evidence bred from this review can be used as input for researchers who intend to make further investigations in this area. The findings of this study could help to achieve the sustainable development goal target of reducing maternal and child mortality.

## Materials and methods

### Data source and searching strategy

Google Scholar, PubMed/Medline, Scopus, Web of Sciences, and grey literature were used to conduct a literature search. All studies published in Ethiopia on pre-eclampsia/hypertensive disorder of pregnancy and its associated factors were included. We also manually searched for cross-references to identify additional relevant articles. The Preferred Reporting Items for Systematic Reviews and Meta-Analyses were used to screen the articles (PRISMA) [20] (**Fig 1**). The search terms and phrases were "Prevalence", "magnitude", "hypertensive disorder of pregnancy", "pre-eclampsia", "determinants", "predictors", "associated factors", "women", and

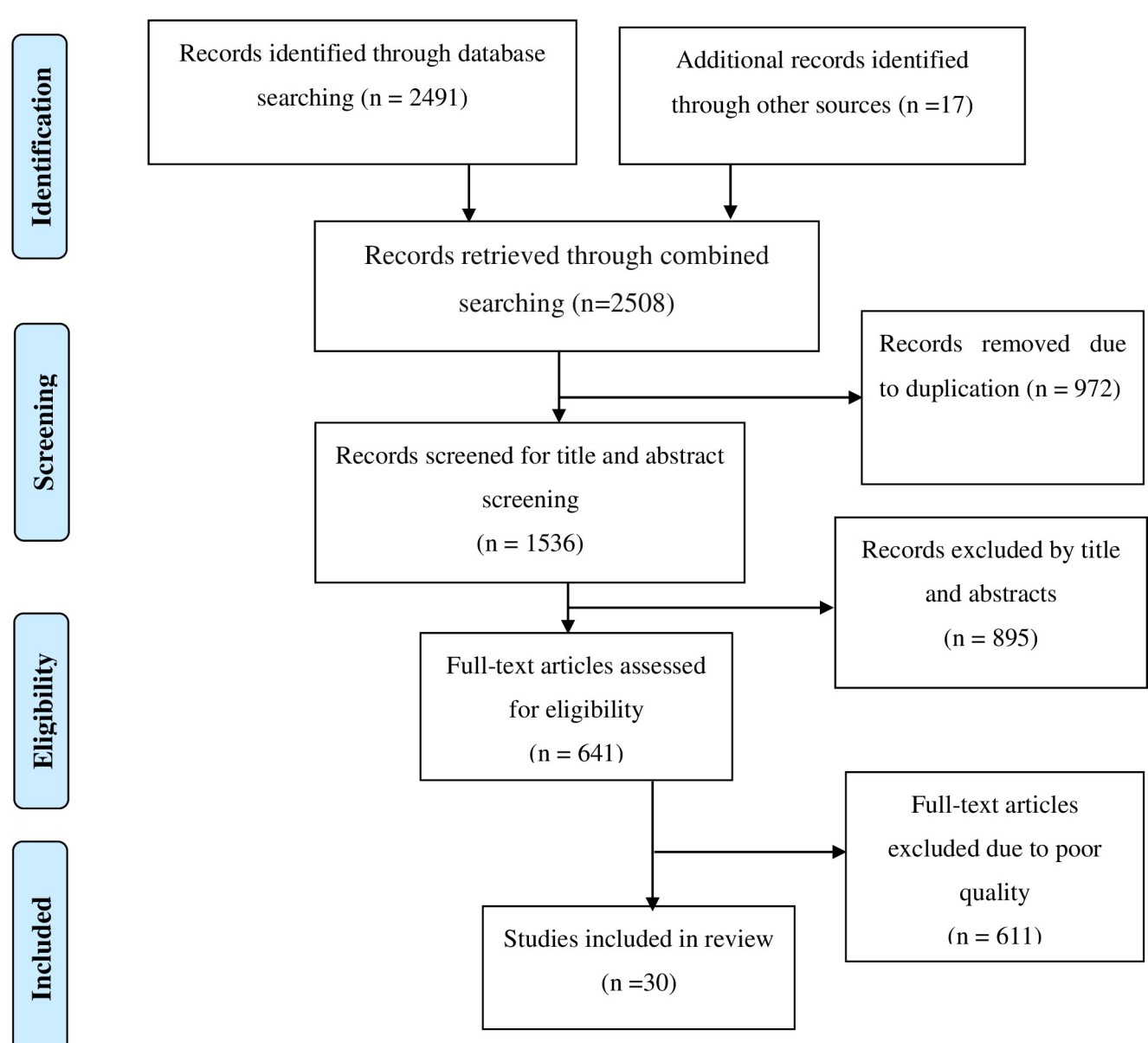

**Fig 1. PRISMA flowchart diagram of the study selection for systematic review and meta-analysis on prevalence of preeclampsia and its determinants in Ethiopia.**

"Ethiopia"). The search strategies were developed using different logical operators (**S1 Table**). Besides this, Ethiopian universities' libraries were used to identify relevant unpublished studies.

### Study selection criteria

**Eligibility criteria.** All published and unpublished observational studies (i.e., cross-sectional, case-control, and cohort designs) that assessed the prevalence of pre-eclampsia and its determinants in Ethiopia, and articles reported in English between January 1, 2013, to January 1, 2023, were included. Meanwhile articles in which the response variable was not clearly defined and measured, pure qualitative papers, studies that were not published in English, case studies, conference papers, editorial notes, systematic reviews, and meta-analyses were excluded.

### Extracted parameters

**Operational definitions.** *Exposures*. Pre-eclampsia risk factors include socio-demographic factors such as age, education, marital status, occupational status, socioeconomic status, alcohol consumption, and wealth status; obstetric factors such as parity, body mass index, age at menarche, history of multiple pregnancy, unwanted pregnancy, gestational diabetes mellitus, history of pre-eclampsia, preterm gestation, and so on; and medical factors (anemia, family history of hypertension, family history of diabetes mellitus, family history of pre-eclampsia, using traditional medicine during pregnancy).

*Outcome*. Pre-eclampsia: new-onset hypertension (BP is >140 mmHg systolic and/or >90 mmHg diastolic) occurring in a pregnant woman after 20 weeks gestation, with proteinuria (defined as urinary excretion of > 0.3g protein in 24 hours it. The primary outcome of this systematic review and meta-analysis was the prevalence of pre-eclampsia and its determinants.

**Selection process and data extraction.** Four authors screened and assessed articles independently (BGK, ADA, AAN, and GNM). Further, we assessed the study's full text based on the aims, methods, participants, and objectives of the findings. Disagreement during data extraction was resolved by discussion and consensus.

Using a Microsoft Excel sheet ((Microsoft Office Professional Plus 2016), all necessary data were extracted from twenty-nine primary studies. This form considers the principal author, year of publication, study setting, sample size, study design, response rate, and prevalence of pre-eclampsia. In addition, an information extraction format was prepared for each specific determinant, i. e., age greater than 35 years, history of hypertension, alcohol intake during pregnancy, primigravida, history of diabetes mellitus, family history of hypertension, family history of diabetes mellitus, housewife and gestational diabetes mellitus. Variables were chosen for this study if they were reported as a significant factor in two or more studies.

**Quality assessment.** Each incorporated study's scientific strength and quality were assessed using the Newcastle-Ottawa Scale quality assessment tool adapted for cross-sectional study quality assessments [21]. Using this tool, four authors (BGK, ADA, AAN, and GNM) independently assessed the quality of each study. The final two authors (BCD and SA) settled disagreements among the four authors. If there were still disagreements among the six authors, the consensus was reached by averaging the six authors' scores. Finally, the quality of the studies was assessed using these criteria; those with medium (satisfying 70% of quality evaluation criteria) and high quality (7 out of 10 scales) were included in the analysis [22] (**S2 Table**).

**Statistical analysis.** We entered data into Microsoft Excel and analyzed it using Stata version 17. Based on the eligibility criteria, the determinants of pre-eclampsia were investigated. We looked at studies that shared at least one determinant factor and a 95% confidence interval (CI). The DerSimonian-Laird random-effects model was used to assess differences between

studies. Text, tables, and forest plots with effect and 95% confidence interval measures were used to present the findings. The $I^2$ statistics were used to test statistical heterogeneity at a p-value of $\leq 0.05$.

**Publication bias and heterogeneity.** To reduce the risk of bias, exhaustive searches were conducted. The authors' collaborative work was also crucial in minimizing bias, selecting articles based on clear objectives and eligibility criteria, determining study quality, and extracting and obtaining information. We investigate publication bias qualitatively by visually inspecting the funnel plot graph [23]. Furthermore, Egger's correlation tests were performed at a 5% significant level to determine the presence of publication bias [24]. Subgroup analysis was also performed by study region, sample size, and study design to decrease the random variations among the primary study's point estimates. A sensitivity analysis was also performed to ascertain the possible cause of heterogeneity. In the random-effect model, heterogeneity across studies was assessed using inverse variance ($I^2$) statistics and the corresponding p-value.

## Results

### Characteristics of the included study

A total of 2508 studies were retrieved by electronic search using PubMed, Scopus, HINARI, web science, and grey literature. Of these articles, 972 duplicates were eliminated while 1536 articles were reviewed further for inclusion. Out of this article, 895 studies were excluded due to irrelevance, and 611 were excluded due to limited statistical analysis, an irrelevant target population, and an inconsistent study report. Finally, the study included twenty-nine articles that met the inclusion criteria, and were included in this systematic review and meta-analysis (**Fig 1**).

A total of twenty-nine qualifying observational studies reported in English were included from five regions in this systematic review and meta-analysis. A total of 31,201 mothers participated in this study. The sample size ranged from 129 in the Oromia region [25] to 8626 in the Amhara region [7]. The prevalence of pre-eclampsia was reported between 2.1% [7] and 49.8% [26] (**Table 1**). Concerning the quality score of the primary studies, all of them had good quality (**S2 Table**).

### Prevalence of pre-eclampsia

Of the studies included, only twenty articles reported the prevalence of pre-eclampsia. Based on the random effect model, the overall pooled prevalence of pre-eclampsia in Ethiopia was 11.51% (95% CI: 8.41, 14.61) with the level of heterogeneity ($I^2 = 99.8\%$, $p < 0.001$ (**Fig 2**).

### Publication bias and heterogeneity

The funnel plot results were asymmetric, indicating the presence of publication bias among the studies included (**Fig 3A**). Egger's regression test revealed the presence of publication bias across studies (p-value $< 0.001$, t = 6.90). The Duval and Tweedie nonparametric trim and fill analysis was used to correct publication bias among the studies. As a result, publication bias was corrected when eight missing studies were filled in the funnel plot using trim and fill analysis. After filling of eight studies, 28 studies were included, and the trim and fill analysis gave the pooled prevalence of pre-eclampsia of 5.31 (1.98–8.63) (**Fig 3B**).

### Sensitivity analysis

To find the potential source of heterogeneity seen in the pooled prevalence of pre-eclampsia, the authors conducted a leave-one-out sensitivity analysis. The result of the sensitivity analysis found that the finding did not rely on a particular study. The pooled prevalence of pre-

**Table 1. Summary of twenty-nine observational studies included in this study assessing prevalence of preeclampsia and determinants in Ethiopia, 2023.**

| S.N | Authors | Year | Regions | Study design | Sample size | Prevalence |
|---|---|---|---|---|---|---|
| 1 | Ayele et al. | 2021 | Amhara | Cross-sectional | 261 | 15.7 |
| 2 | Birhanu et al. | 2020 | Amhara | Cohort | 242 | 3.35 |
| 3 | Tessema et al. | 2015 | Amhara | Cross-sectional | 490 | 8.4 |
| 4 | Ayalew et al. | 2019 | Amhara | Cross-sectional | 193 | 13 |
| 5 | Vata et al. | 2015 | SNNPR | Cohort | 172 | 2.23 |
| 6 | Mareg et al. | 2020 | SNNPR | Case-control | 240 | 33.3 |
| 7 | Demissie Beketie et al. | 2022 | SNNPR | Case-control | 426 | 33.3 |
| 8 | Andarge RB et al. | 2020 | SNNPR | Cross-sectional | 242 | 9.9 |
| 9 | Belay & Wudad | 2019 | Oromia | Cross-sectional | 129 | 12.4 |
| 10 | Fikadu et al. | 2020 | SNNPR | Case-control | 527 | 33.3 |
| 11 | Birhanu Jikamo et al. | 2022 | SNNPR | Case-control | 816 | 30.94 |
| 12 | Haile et al. | 2021 | Tigray | Case-control | 344 | 33.3 |
| 13 | Asres et al | 2022 | SNNPR | Case-control | 1065 | 33.3 |
| | Fantahunegne | 2019 | Amhara | Case-control | 291 | 33.3 |
| 14 | Shegaze et al. | 2016 | SNNPR | Cross-sectional | 422 | 18.25 |
| 15 | Grum et al. | 2017 | Addis Ababa | Case-control | 291 | 33.3 |
| 16 | Wagnew et al. | 2016 | Addis Ababa | Cross-sectional | 1809 | 4.2 |
| 17 | Mohammed et al. | 2017 | Addis Ababa | Case-control | 261 | 33.3 |
| 18 | Katore et al. | 2021 | Oromia | Case-control | 306 | 33.3 |
| 19 | Hinkosa et al. | 2020 | Oromia | Cross-sectional | 6826 | 2.24 |
| 20 | Legesse et al. | 2019 | Tigray | Cross-sectional | 502 | 4.3 |
| 21 | H. B. Kahsay, and F. E. Gashe | 2018 | Tigray | Case-control | 1347 | 10.6 |
| 22 | Wodajo & Reddy | 2016 | Amhara | Cross-sectional | 320 | 6.6 |
| 23 | T. A. Gudeta and T. M. Regassa | 2019 | SNNPR | Cross-sectional | 422 | 6.4 |
| 24 | Gudeta et al. | 2018 | Oromia | Cross-sectional | 356 | 6.5 |
| 25 | Terefe et al. | 2015 | Amhara | Cross-sectional | 8626 | 2.1 |
| 26 | Asfaw | 2014 | Addis Ababa | Cross-sectional | 3351 | 7.8 |
| 27 | Mekie et al. | 2020 | Amhara | Case-control | 330 | 33.3 |
| 28 | Belayhun et al. | 2021 | SNNPR | Case-control | 283 | 16.7 |
| 29 | Alemie | 2021 | Amhara | Cross-sectional | 311 | 49.8 |

eclampsia varied and ranged from 9.49% (7.13, 11.86) to 12.01% (8.69, 15.33) after the insertion of eight studies (**Fig 4**).

## Subgroup analysis

Subgroup analysis was conducted by region, design, and sample size, and revealed substantial heterogeneity across the studies on prevalence of preeclampsia ($I^2 > 78.6\%$, P = 0.03). Prevalence of pre-eclampsia by region showed highest in the Amhara region; 14.08 (6.7–21.47) and lowest in Addis Ababa; 5.98 (2.45–9.51). Moreover, the result of subgroup analysis by study design and sample size of the highest prevalence of pre-eclampsia showed that 19.39 (8.38–30.4) among case-control and 12.97 (8.51–17.43) sample size less or equal to five thundered respectively (**Figs 5–7**).

## Determinants of pre-eclampsia

In the current systematic review and meta-analysis; maternal age, housewife, drinking alcohol during pregnancy, history of preeclampsia, history of GDM, primigravida, family history of

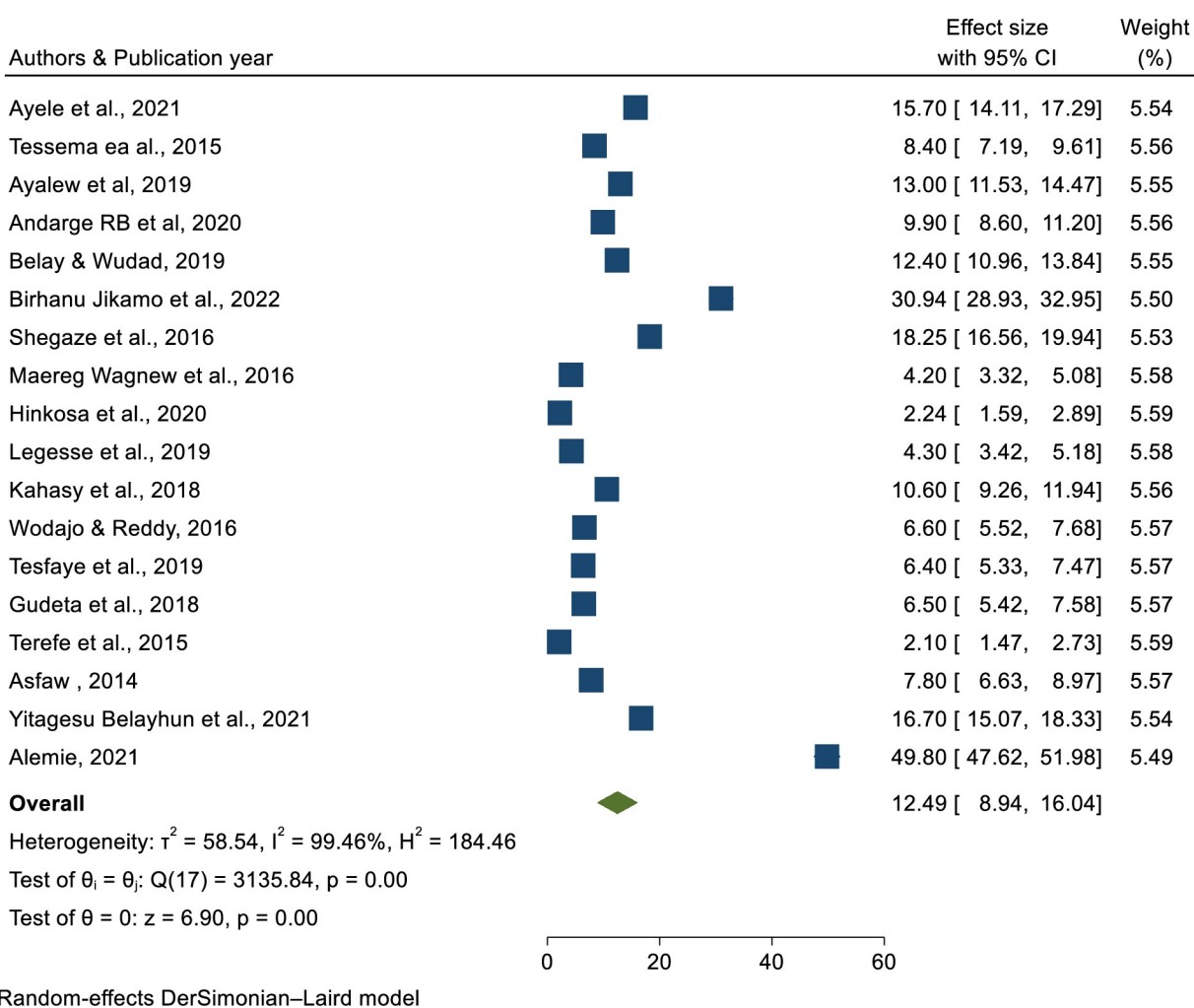

**Fig 2. Forest plot of the pooled prevalence of preeclampsia in Ethiopia, 2023.**

hypertension, family history of DM, history of chronic hypertension, history of multiple pregnancies, BMI $\geq$ 30 kg/m$^2$ and history of DM were found to be determinants of pre-eclampsia.

### Association between maternal age and pre-eclampsia

Two studies were included; those with a maternal age less than or equal to 24 years were associated with pre-eclampsia [25, 27]. Only one of the included studies found pre-eclampsia determinants, but the final pooled meta-regression analysis found not statistically significant for pre-eclampsia (AOR = 1.5, 95% CI: -0.46, 3.46). This meta-analysis had a high degree of heterogeneity was observed, with I$^2$ = 99.22% and a P-value of $\leq$ 0.001. Three studies, however, described maternal ages greater than 35 years [28–30]. Three of them had a strong link to pre-eclampsia. In addition, a pooled meta-regression analysis revealed that pre-eclampsia was statistically significant (AOR = 2.34, 95% CI: 1.74, 2.94). There is no observed heterogeneity across the studies (**Fig 8**).

### Association of gravidity and pre-eclampsia

This subcategory had six articles. Three of them are statistically associated with pre-eclampsia [31–33], but the remaining three are not [27, 29, 34]. According to the pooled meta-regression

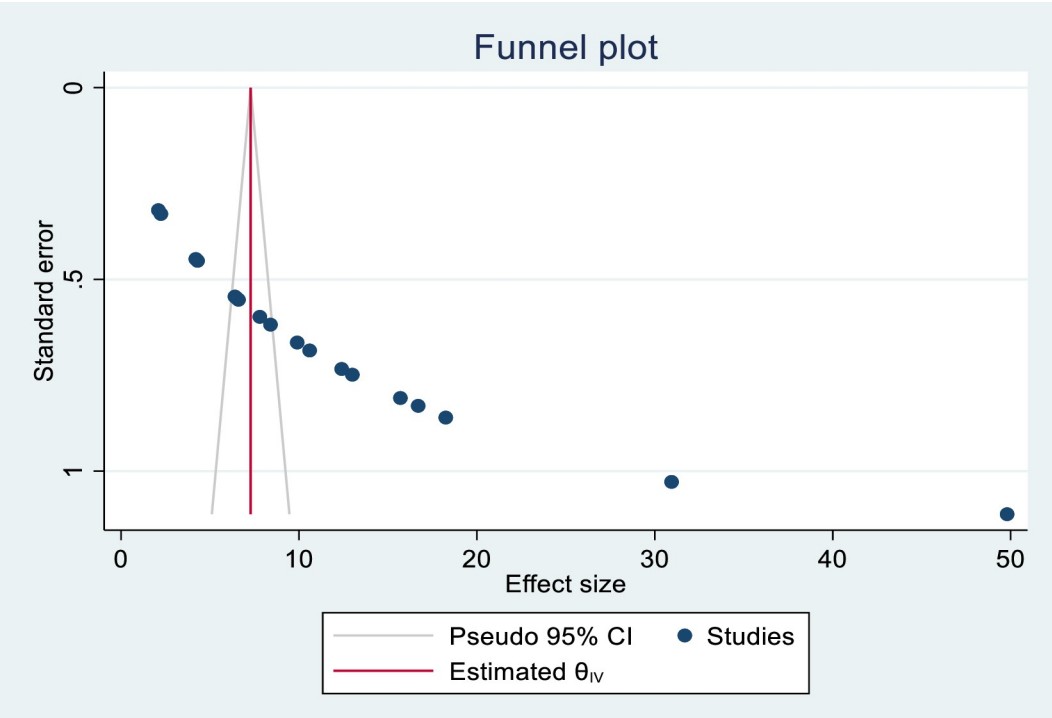

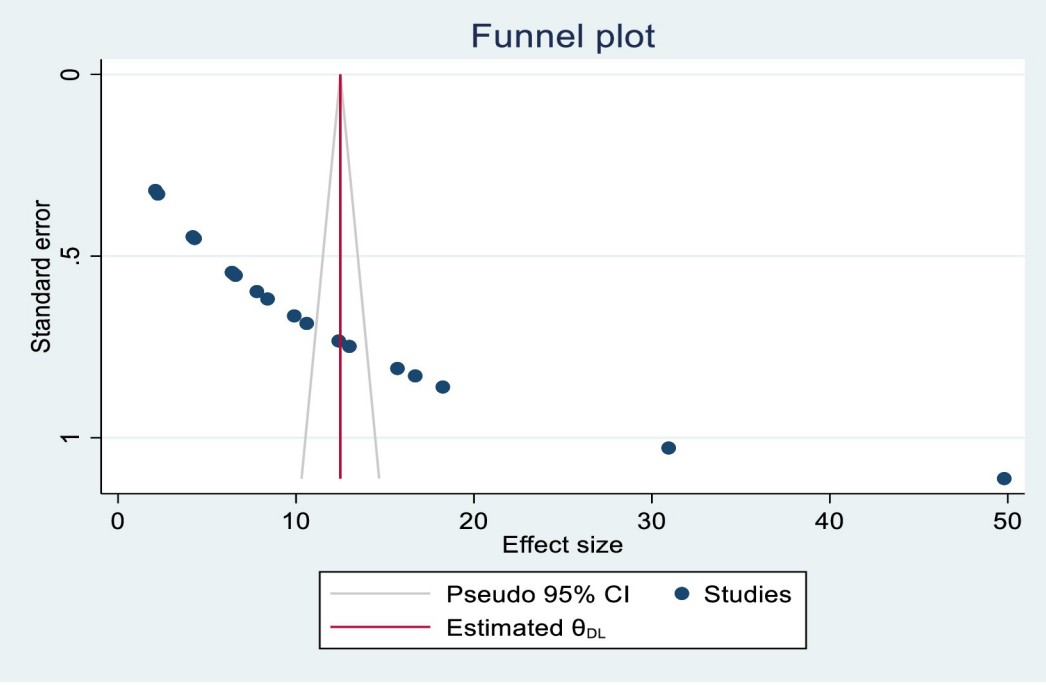

**Fig 3.** (A) Funnel plot to test publication bias of eighteen studies. (B) Result of trim and fill analysis for adjusting publication bias of the eighteen studies (unchanged).

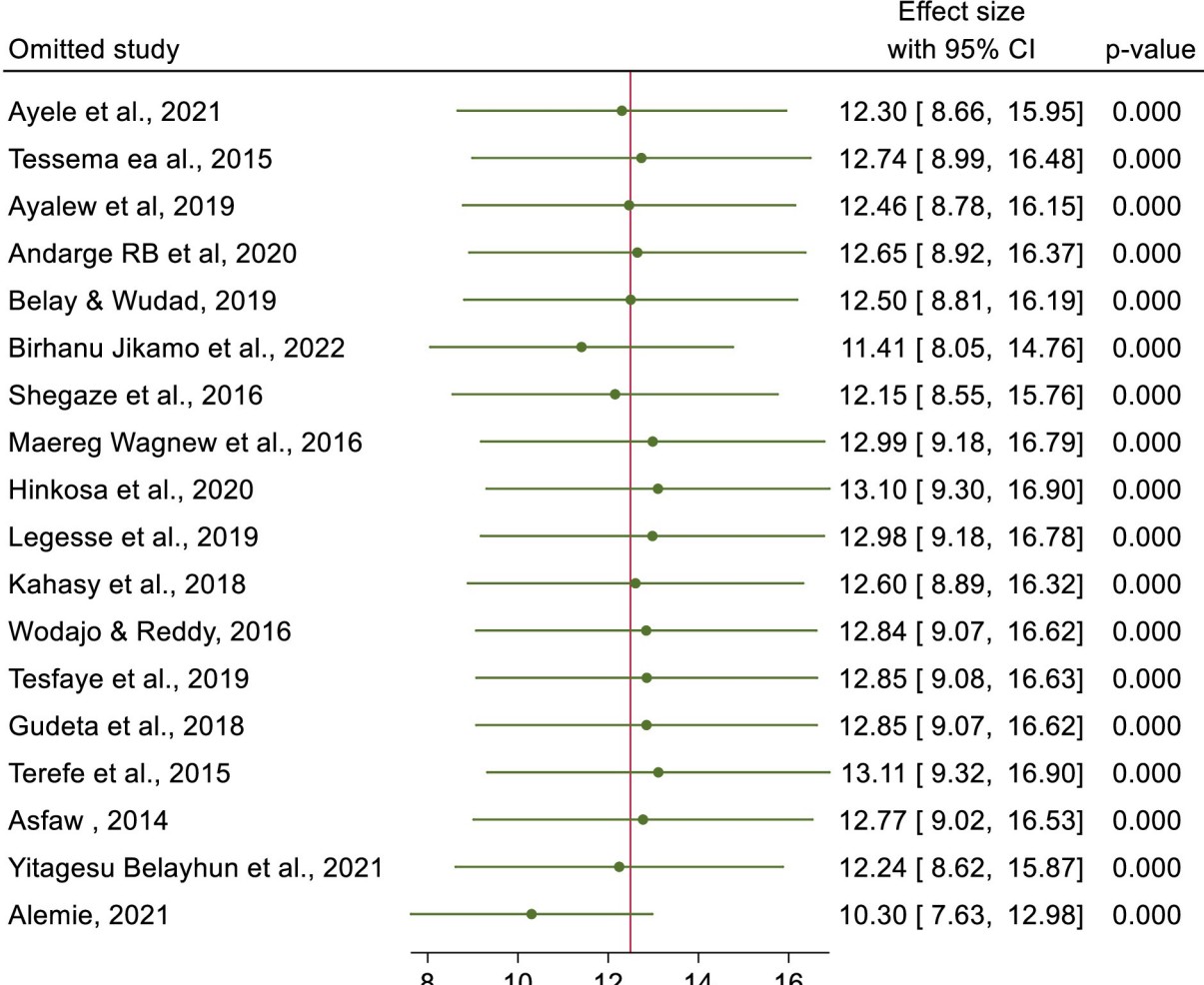

| Omitted study | | Effect size with 95% CI | p-value |
|---|---|---|---|
| Ayele et al., 2021 | | 12.30 [ 8.66, 15.95] | 0.000 |
| Tessema ea al., 2015 | | 12.74 [ 8.99, 16.48] | 0.000 |
| Ayalew et al, 2019 | | 12.46 [ 8.78, 16.15] | 0.000 |
| Andarge RB et al, 2020 | | 12.65 [ 8.92, 16.37] | 0.000 |
| Belay & Wudad, 2019 | | 12.50 [ 8.81, 16.19] | 0.000 |
| Birhanu Jikamo et al., 2022 | | 11.41 [ 8.05, 14.76] | 0.000 |
| Shegaze et al., 2016 | | 12.15 [ 8.55, 15.76] | 0.000 |
| Maereg Wagnew et al., 2016 | | 12.99 [ 9.18, 16.79] | 0.000 |
| Hinkosa et al., 2020 | | 13.10 [ 9.30, 16.90] | 0.000 |
| Legesse et al., 2019 | | 12.98 [ 9.18, 16.78] | 0.000 |
| Kahasy et al., 2018 | | 12.60 [ 8.89, 16.32] | 0.000 |
| Wodajo & Reddy, 2016 | | 12.84 [ 9.07, 16.62] | 0.000 |
| Tesfaye et al., 2019 | | 12.85 [ 9.08, 16.63] | 0.000 |
| Gudeta et al., 2018 | | 12.85 [ 9.07, 16.62] | 0.000 |
| Terefe et al., 2015 | | 13.11 [ 9.32, 16.90] | 0.000 |
| Asfaw , 2014 | | 12.77 [ 9.02, 16.53] | 0.000 |
| Yitagesu Belayhun et al., 2021 | | 12.24 [ 8.62, 15.87] | 0.000 |
| Alemie, 2021 | | 10.30 [ 7.63, 12.98] | 0.000 |

Random-effects DerSimonian–Laird model

**Fig 4. Sensitivity analysis of the prevalence of preeclampsia and its determinants in Ethiopia, 2023.**

analysis, there is no statistically significant difference in the occurrence of pre-eclampsia in primigravida (AOR = 1.76, 95%CI: 0. 65, 2.87). Because the heterogeneity test revealed more heterogeneity across studies, we used the random-effects model to calculate the pooled odds ratio (**Fig 8**).

## Association of occupation and pre-eclampsia

According to two primary articles, being a housewife is another important determinant of pre-eclampsia [32, 35]. As a result, the final pooled odds ratio was calculated using random-effect model analysis and was found to be significantly associated with pre-eclampsia (AOR = 2.76, 95%CI: 1. 20, 4.32). There is no evidence of heterogeneity across studies, as proved by ($I^2$ = 0.00%, p = 0.37) (**Fig 8**).

## Association of alcohol intake and pre-eclampsia

Five primary studies found that alcohol consumption during pregnancy was a risk factor for preeclampsia. Three studies found no statistically significant effects on pre-eclampsia [4, 33, 36], but two of them were statistically significant with pre-eclampsia [31, 37]. The

|  | | Effect size with 95% CI | Weight (%) |
|---|---|---|---|
| Study | | | |
| **Addis Ababa** | | | |
| Maereg Wagnew et al., 2016 | | 4.20 [ 3.32, 5.08] | 5.58 |
| Asfaw , 2014 | | 7.80 [ 6.63, 8.97] | 5.57 |
| Heterogeneity: $\tau^2$ = 6.20, $I^2$ = 95.70%, $H^2$ = 23.28 | | 5.98 [ 2.45, 9.51] | |
| Test of $\theta_i = \theta_j$: Q(1) = 23.28, p = 0.00 | | | |
| **Amhara** | | | |
| Ayele et al., 2021 | | 15.70 [ 14.11, 17.29] | 5.54 |
| Tessema ea al., 2015 | | 8.40 [ 7.19, 9.61] | 5.56 |
| Ayalew et al, 2019 | | 13.00 [ 11.53, 14.47] | 5.55 |
| Wodajo & Reddy, 2016 | | 6.60 [ 5.52, 7.68] | 5.57 |
| Terefe et al., 2015 | | 2.10 [ 1.47, 2.73] | 5.59 |
| Alemie, 2021 | | 49.80 [ 47.62, 51.98] | 5.49 |
| Heterogeneity: $\tau^2$ = 137.19, $I^2$ = 99.74%, $H^2$ = 378.65 | | 15.89 [ 6.50, 25.28] | |
| Test of $\theta_i = \theta_j$: Q(5) = 1893.27, p = 0.00 | | | |
| **Oromia** | | | |
| Belay & Wudad, 2019 | | 12.40 [ 10.96, 13.84] | 5.55 |
| Hinkosa et al., 2020 | | 2.24 [ 1.59, 2.89] | 5.59 |
| Gudeta et al., 2018 | | 6.50 [ 5.42, 7.58] | 5.57 |
| Heterogeneity: $\tau^2$ = 23.28, $I^2$ = 98.87%, $H^2$ = 88.19 | | 7.02 [ 1.52, 12.51] | |
| Test of $\theta_i = \theta_j$: Q(2) = 176.38, p = 0.00 | | | |
| **SNNPR** | | | |
| Andarge RB et al, 2020 | | 9.90 [ 8.60, 11.20] | 5.56 |
| Birhanu Jikamo et al., 2022 | | 30.94 [ 28.93, 32.95] | 5.50 |
| Shegaze et al., 2016 | | 18.25 [ 16.56, 19.94] | 5.53 |
| Tesfaye et al., 2019 | | 6.40 [ 5.33, 7.47] | 5.57 |
| Yitagesu Belayhun et al., 2021 | | 16.70 [ 15.07, 18.33] | 5.54 |
| Heterogeneity: $\tau^2$ = 73.98, $I^2$ = 99.25%, $H^2$ = 132.57 | | 16.41 [ 8.84, 23.98] | |
| Test of $\theta_i = \theta_j$: Q(4) = 530.27, p = 0.00 | | | |
| **Tigray** | | | |
| Legesse et al., 2019 | | 4.30 [ 3.42, 5.18] | 5.58 |
| Kahasy et al., 2018 | | 10.60 [ 9.26, 11.94] | 5.56 |
| Heterogeneity: $\tau^2$ = 19.51, $I^2$ = 98.30%, $H^2$ = 58.94 | | 7.43 [ 1.26, 13.60] | |
| Test of $\theta_i = \theta_j$: Q(1) = 58.94, p = 0.00 | | | |
| **Overall** | | 12.49 [ 8.94, 16.04] | |
| Heterogeneity: $\tau^2$ = 58.54, $I^2$ = 99.46%, $H^2$ = 184.46 | | | |
| Test of $\theta_i = \theta_j$: Q(17) = 3135.84, p = 0.00 | | | |
| Test of group differences: $Q_b$(4) = 8.85, p = 0.06 | | | |

Random-effects DerSimonian–Laird model

**Fig 5. Sub-group analysis of the prevalence of preeclampsia in Ethiopia by region, 2023.**

| Study | Effect size with 95% CI | Weight (%) |
|---|---|---|
| **Case-control** | | |
| Birhanu Jikamo et al., 2022 | 30.94 [ 28.93, 32.95] | 5.50 |
| Kahasy et al., 2018 | 10.60 [ 9.26, 11.94] | 5.56 |
| Yitagesu Belayhun et al., 2021 | 16.70 [ 15.07, 18.33] | 5.54 |
| Heterogeneity: $\tau^2$ = 93.97, $I^2$ = 99.26%, $H^2$ = 135.54 | 19.39 [ 8.38, 30.40] | |
| Test of $\theta_i = \theta_j$: Q(2) = 271.08, p = 0.00 | | |
| | | |
| **Cross-sectional** | | |
| Ayele et al., 2021 | 15.70 [ 14.11, 17.29] | 5.54 |
| Tessema ea al., 2015 | 8.40 [ 7.19, 9.61] | 5.56 |
| Ayalew et al, 2019 | 13.00 [ 11.53, 14.47] | 5.55 |
| Andarge RB et al, 2020 | 9.90 [ 8.60, 11.20] | 5.56 |
| Belay & Wudad, 2019 | 12.40 [ 10.96, 13.84] | 5.55 |
| Shegaze et al., 2016 | 18.25 [ 16.56, 19.94] | 5.53 |
| Maereg Wagnew et al., 2016 | 4.20 [ 3.32, 5.08] | 5.58 |
| Hinkosa et al., 2020 | 2.24 [ 1.59, 2.89] | 5.59 |
| Legesse et al., 2019 | 4.30 [ 3.42, 5.18] | 5.58 |
| Wodajo & Reddy, 2016 | 6.60 [ 5.52, 7.68] | 5.57 |
| Tesfaye et al., 2019 | 6.40 [ 5.33, 7.47] | 5.57 |
| Gudeta et al., 2018 | 6.50 [ 5.42, 7.58] | 5.57 |
| Terefe et al., 2015 | 2.10 [ 1.47, 2.73] | 5.59 |
| Asfaw , 2014 | 7.80 [ 6.63, 8.97] | 5.57 |
| Alemie, 2021 | 49.80 [ 47.62, 51.98] | 5.49 |
| Heterogeneity: $\tau^2$ = 49.65, $I^2$ = 99.42%, $H^2$ = 172.82 | 11.11 [ 7.53, 14.69] | |
| Test of $\theta_i = \theta_j$: Q(14) = 2419.43, p = 0.00 | | |
| | | |
| **Overall** | 12.49 [ 8.94, 16.04] | |
| Heterogeneity: $\tau^2$ = 58.54, $I^2$ = 99.46%, $H^2$ = 184.46 | | |
| Test of $\theta_i = \theta_j$: Q(17) = 3135.84, p = 0.00 | | |
| | | |
| Test of group differences: $Q_b$(1) = 1.96, p = 0.16 | | |

Random-effects DerSimonian–Laird model

**Fig 6. Sub-group analysis of the prevalence of preeclampsia in Ethiopia by study design, 2023.**

heterogeneity test showed moderate heterogeneity across the studies ($I^2$ = 51.39%, p-value = 0.08); as a result, we used the random-effects model to estimate the pooled odds ratio of determinants of pre-eclampsia (AOR = 1.53, 95% CI = 1.03, 2.04) (**Fig 9**).

## Association of Body Mass index and pre-eclampsia

Two articles have been included in this categorization [4, 27]. Only one study found a statistically significant link between pre-eclampsia and BMI $\geq$ 30 kg/m$^2$. However, the pooled meta-

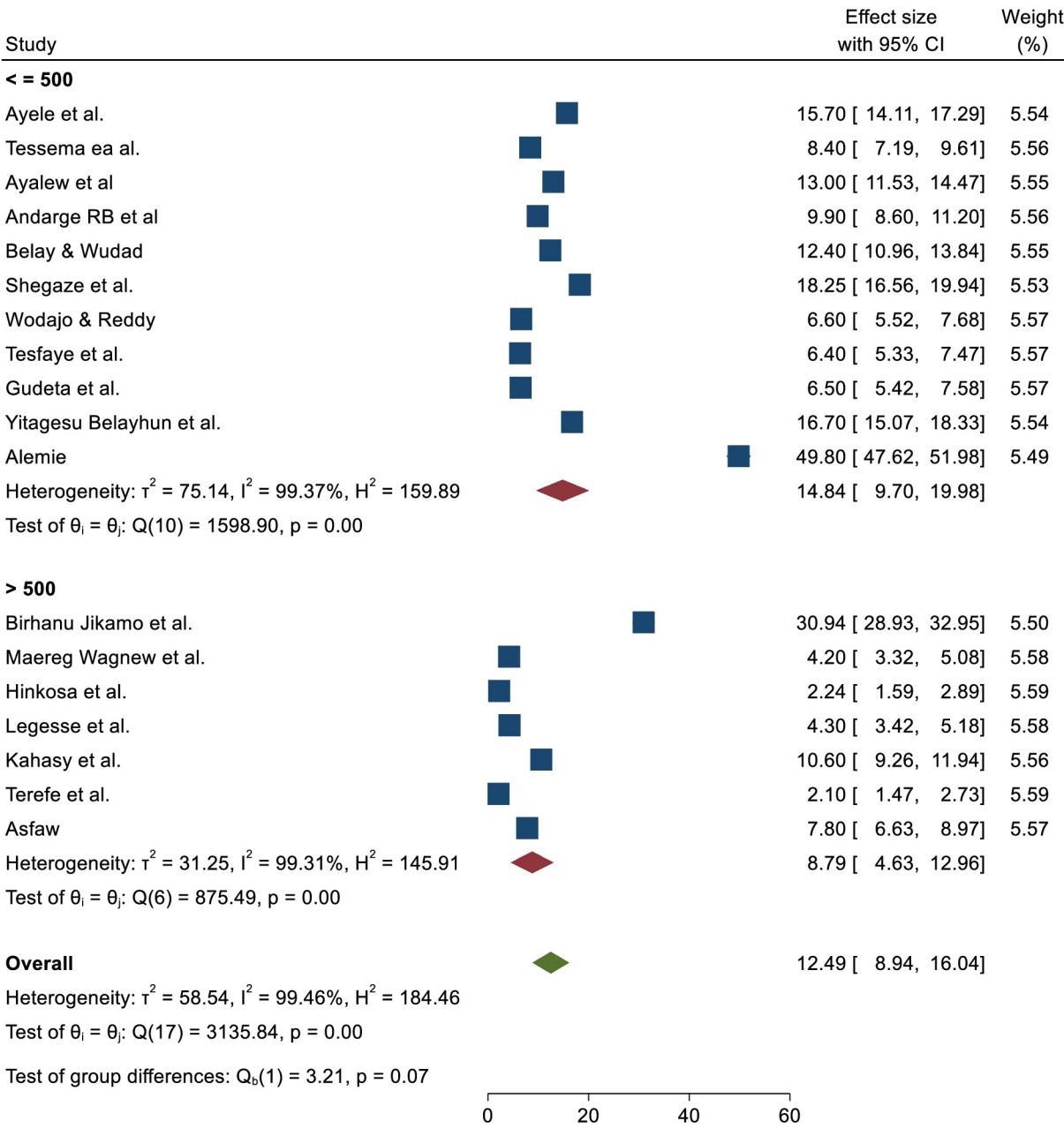

| Study | Effect size with 95% CI | Weight (%) |
|---|---|---|
| **< = 500** | | |
| Ayele et al. | 15.70 [ 14.11, 17.29] | 5.54 |
| Tessema ea al. | 8.40 [ 7.19, 9.61] | 5.56 |
| Ayalew et al | 13.00 [ 11.53, 14.47] | 5.55 |
| Andarge RB et al | 9.90 [ 8.60, 11.20] | 5.56 |
| Belay & Wudad | 12.40 [ 10.96, 13.84] | 5.55 |
| Shegaze et al. | 18.25 [ 16.56, 19.94] | 5.53 |
| Wodajo & Reddy | 6.60 [ 5.52, 7.68] | 5.57 |
| Tesfaye et al. | 6.40 [ 5.33, 7.47] | 5.57 |
| Gudeta et al. | 6.50 [ 5.42, 7.58] | 5.57 |
| Yitagesu Belayhun et al. | 16.70 [ 15.07, 18.33] | 5.54 |
| Alemie | 49.80 [ 47.62, 51.98] | 5.49 |
| Heterogeneity: $\tau^2$ = 75.14, $I^2$ = 99.37%, $H^2$ = 159.89 | 14.84 [ 9.70, 19.98] | |
| Test of $\theta_i = \theta_j$: Q(10) = 1598.90, p = 0.00 | | |
| | | |
| **> 500** | | |
| Birhanu Jikamo et al. | 30.94 [ 28.93, 32.95] | 5.50 |
| Maereg Wagnew et al. | 4.20 [ 3.32, 5.08] | 5.58 |
| Hinkosa et al. | 2.24 [ 1.59, 2.89] | 5.59 |
| Legesse et al. | 4.30 [ 3.42, 5.18] | 5.58 |
| Kahasy et al. | 10.60 [ 9.26, 11.94] | 5.56 |
| Terefe et al. | 2.10 [ 1.47, 2.73] | 5.59 |
| Asfaw | 7.80 [ 6.63, 8.97] | 5.57 |
| Heterogeneity: $\tau^2$ = 31.25, $I^2$ = 99.31%, $H^2$ = 145.91 | 8.79 [ 4.63, 12.96] | |
| Test of $\theta_i = \theta_j$: Q(6) = 875.49, p = 0.00 | | |
| | | |
| **Overall** | 12.49 [ 8.94, 16.04] | |
| Heterogeneity: $\tau^2$ = 58.54, $I^2$ = 99.46%, $H^2$ = 184.46 | | |
| Test of $\theta_i = \theta_j$: Q(17) = 3135.84, p = 0.00 | | |
| Test of group differences: $Q_b(1)$ = 3.21, p = 0.07 | | |

Random-effects DerSimonian–Laird model

**Fig 7. Sub-group analysis of the prevalence of preeclampsia in Ethiopia by sample size, 2023.**

regression analysis revealed no statistically significant link between BMI and pre-eclampsia (AOR = 1.52 (95% CI = 0.56, 2.58). $I^2$ revealed that the studies included in this analysis were heterogeneous ($I^2$ = 72.73%, p-value = 0.06) (**Fig 9**).

## Association of history of chronic hypertension with pre-eclampsia

According to two primary studies, a history of chronic hypertension is another important determinant of pre-eclampsia [4, 30]. Those mothers having a history of chronic hypertension

**Fig 8. Determinants of preeclampsia in Ethiopia, 2023.**

were nearly two times (AOR = 2.44, 95% CI = 1.8, 3.08) increased odds of developing pre-eclampsia than their counterparts. The heterogeneity test showed there is no heterogeneity across the studies ($I^2$ = 0.00%, p = 0.67) (**Fig 9**).

|  | Effect size with 95% CI | Weight (%) |
|---|---|---|
| **Study** | | |
| **Alcohol drinking during pregnancy** | | |
| Ayele et al., 2021 | 0.89 [ 0.18,  1.61] | 8.02 |
| Demissie Beketie et al., 2022 | 3.06 [ 1.62,  4.49] | 3.20 |
| Haile et al., 2021 | 1.61 [ 0.97,  2.25] | 8.92 |
| Shegaze et al., 2016 | 1.18 [ 0.22,  2.13] | 5.78 |
| Grum et al., 2017 | 1.69 [ 1.09,  2.28] | 9.49 |
| Heterogeneity: $\tau^2 = 0.17$, $I^2 = 51.39\%$, $H^2 = 2.06$ | 1.53 [ 1.03,  2.04] | |
| Test of $\theta_i = \theta_j$: Q(4) = 8.23, p = 0.08 | | |
| **BMI > 30 kg/m2** | | |
| Ayele et al., 2021 | 1.09 [ 0.46,  1.73] | 8.92 |
| Mohammed et al., 2017 | 2.12 [ 1.28,  2.96] | 6.73 |
| Heterogeneity: $\tau^2 = 0.39$, $I^2 = 72.73\%$, $H^2 = 3.67$ | 1.57 [ 0.56,  2.58] | |
| Test of $\theta_i = \theta_j$: Q(1) = 3.67, p = 0.06 | | |
| **History of chronic hypertension** | | |
| Ayele et al., 2021 | 2.58 [ 1.67,  3.50] | 6.09 |
| Tessema  et al., 2015 | 2.30 [ 1.40,  3.20] | 6.23 |
| Heterogeneity: $\tau^2 = 0.00$, $I^2 = 0.00\%$, $H^2 = 1.00$ | 2.44 [ 1.80,  3.08] | |
| Test of $\theta_i = \theta_j$: Q(1) = 0.18, p = 0.67 | | |
| **History of multiple pregnancy** | | |
| Birhanu et al., 2020 | 1.68 [ 0.02,  3.35] | 2.51 |
| Demissie Beketie et al., 2022 | 1.09 [ 0.46,  1.73] | 8.92 |
| Andarge RB et al., 2020 | 2.12 [ 1.28,  2.96] | 6.73 |
| Belay & Wudad, 2019 | 2.18 [ 0.49,  3.88] | 2.44 |
| Asres et al., 2022 | 1.10 [ 0.34,  1.86] | 7.53 |
| Grum et al., 2017 | 1.50 [ 0.83,  2.18] | 8.48 |
| Heterogeneity: $\tau^2 = 0.01$, $I^2 = 5.90\%$, $H^2 = 1.06$ | 1.45 [ 1.09,  1.80] | |
| Test of $\theta_i = \theta_j$: Q(5) = 5.31, p = 0.38 | | |
| **Overall** | 1.63 [ 1.34,  1.92] | |
| Heterogeneity: $\tau^2 = 0.14$, $I^2 = 44.76\%$, $H^2 = 1.81$ | | |
| Test of $\theta_i = \theta_j$: Q(14) = 25.35, p = 0.03 | | |
| Test of group differences: $Q_b(3) = 7.29$, p = 0.06 | | |

0                                 5

Random-effects DerSimonian–Laird model

**Fig 9. Determinants of preeclampsia in Ethiopia, 2023.**

## Association of history of multiple pregnancy and pre-eclampsia

This meta-analysis included six studies. Only one of the included studies found a higher risk of pre-eclampsia in women with a history of multiple pregnancies [38]. While five studies found no difference in the development of pre-eclampsia whether mothers had a history of multiple pregnancies or not [25, 28, 31, 34, 37]. The final pooled meta-regression analysis showed statistically significant difference in the development of pre-eclampsia between those had or not have a history of multiple pregnancies (AOR = 1.45, 95% CI = 1.09, 1.8). In this meta-analysis category, no heterogeneity was found as shown by ($I^2$ = 5.9%, P-value = 0.38) (**Fig 9**).

## Association of family history of diabetes mellitus and pre-eclampsia

Four studies were included in this category of meta-analysis. None of them showed a significant relationship with pre-eclampsia [30, 34, 35, 39]. The final pooled analysis also revealed that there was no statistically significant difference in the occurrence of pre-eclampsia between mothers who had a family history of diabetes mellitus and those who did not (AOR = 0.9, 95% CI = 0.54, 1.26). The heterogeneity test revealed that there was moderate heterogeneity in the studies ($I^2$ = 31.94%, P-value = 0.22) (**Fig 10**).

## Association of family history of hypertension and pre-eclampsia

Nine studies were included in this category of meta-analysis to assess a family history of hypertension as a risk factor for pre-eclampsia [6, 27, 29, 30, 33–37]. Eight of the included studies found no statistically significant difference between mothers who had a family history of hypertension and those who did not. The pooled meta-regression analysis revealed a statistically significant difference in the occurrence of pre-eclampsia with a family history of hypertension (AOR = 1.84, 95% CI = 1.39, 2.3). The heterogeneity test ($I^2$ = 3.76, p-value = 0.40) showed that, no heterogeneity within the studies (**Fig 10**).

## Association of gestational diabetes mellitus and pre-eclampsia

Two studies were included in this specific category of meta-analysis to assess the history of gestational diabetes mellitus as a risk factor for pre-eclampsia [6, 38] One study found a statistically significant link between a history of gestational diabetes mellitus and pre-eclampsia. One study, however, found no significant link with preeclampsia. The pooled meta-regression analysis showed that there is not statistically significant association between the occurrence of pre-eclampsia and women having of history of gestational diabetes mellitus (AOR = 1.41, 95% CI = 0.39, 2.43). There was higher heterogeneity between the studies ($I^2$ = 91.44%, p < 0.001). As a result, the pooled odds ratio was estimated using a random-effect model (**Fig 10**).

## Association of history of diabetes mellitus and pre-eclampsia

Four studies were included in the review of diabetes mellitus history as a risk factor for pre-eclampsia [25, 28, 32, 33]. Three of the included studies showed a statistically significant association between a history of diabetes mellitus with pre-eclampsia [25, 28, 33]. While one study did not show a significant association with pre-eclampsia [32]. The pooled meta-regression analysis showed that there is a statistically significant association between the occurrence of pre-eclampsia and women having a history of diabetes mellitus (AOR = 2.02, 95%CI, 1.58, 2.47). The heterogeneity test showed that (I2 = 6.67%, p-value = 0.36); no heterogeneity within the studies (**Fig 10**).

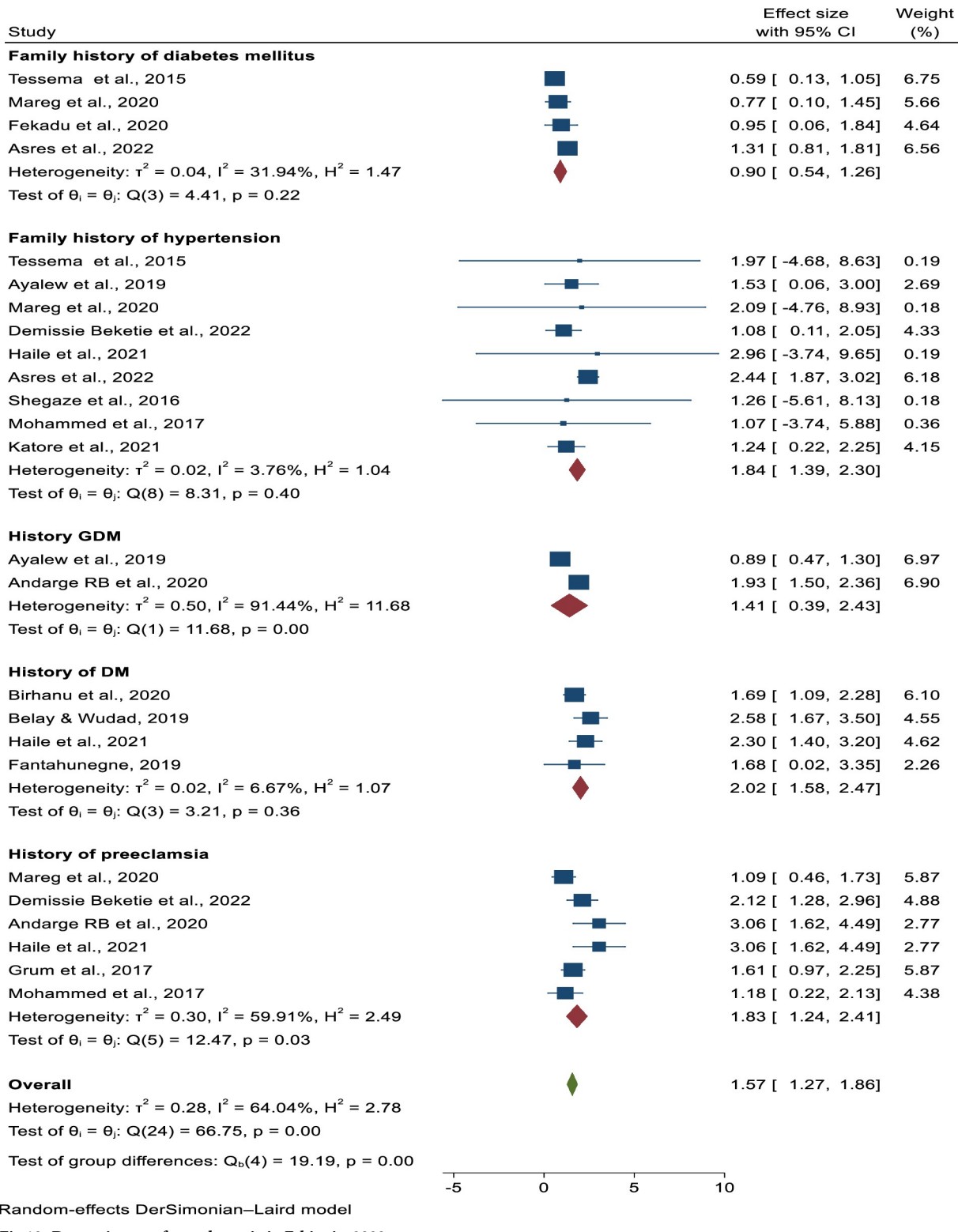

**Fig 10. Determinants of preeclampsia in Ethiopia, 2023.**

## Association of previous history of pre-eclampsia and pre-eclampsia

Six studies were included in this category of meta-analysis to assess previous history of preeclampsia as a risk factor for pre-eclampsia [27, 31, 33, 35, 37, 38]. Three of the included studies had shown a statistically significant association between previous history of pre-eclampsia and pre-eclampsia [33, 37, 38]. However, three studies showed no significant association with pre-eclampsia [27, 31, 35]. The pooled meta-regression analysis showed that there is a statistically significant association in the occurrence of pre-eclampsia and previous history of pre-eclampsia (AOR = 1.83, 95%CI = 1.24, 2.41). The heterogeneity test showed that there is a presence of moderate heterogeneity within the included studies shown by $I^2$ = 59.91%, p-value = 0.03) (**Fig 10**).

## Discussion

The purpose of this systematic review and meta-analysis was conducted to show the pooled prevalence and determinants of pre-eclampsia in Ethiopia. The pooled prevalence of pre-eclampsia in this study was 11.51% with 95% CI (8.41–14.61). This finding was slightly higher than the previous meta-analysis conducted by Tesfa and his colleagues (4.74%) [40], Berhe and his colleagues (5.47%) [41], Gemechu and his colleagues (4.1%) [42], and WHO multicounty survey (2.16%) [43]. Therefore, it should be focused on prevention strategies for pre-eclampsia to reduce the incidence of pre-eclampsia. However, the pooled prevalence of pre-eclampsia in this review was lower than in the meta-analysis study conducted in Africa, which was 22.1% versus 44% [44], and the survey study conducted in China, 19.59% [45]. This disparity could be attributed to ethnic differences, the number of studies including the study setting, the study design, the study population, and sociocultural differences among study participants. Besides that, all the studies included in this meta-analysis were conducted in healthcare facilities, which could have increased the prevalence of pre-eclampsia.

The highest and lowest pooled prevalence of pre-eclampsia seen in Amhara region and Addis Ababa city, respectively, was 14.08% with 95% CI (6.7–21.47) and 5.98% with 95% CI (2.45–9.51). This disparity could be attributed to differences in study design and outcome of interest for the assessment of pre-eclampsia determinants in Ethiopia.

According to the findings of this meta-analysis, there is a significant association between pre-eclampsia and increasing age. Women aged more than 35 years old were two times more likely to develop pre-eclampsia than women aged less than 24 years old (AOR = 2.34, 95% CI: 1.74, 2.94). This is consistent with the findings of studies conducted on women in Kenya, China, Asia, Latin America, and the Caribbean [45, 46]. The increased risk of in older mothers may be due to an abnormally elevated lipid profile, high-density lipid cholesterol, and a higher risk of vascular damage in this age group in comparison to younger women [2, 47]. There was also a 4% increase in the rate of late pre-eclampsia and gestational hypertension for every year over the age of 32 [9]. Early screening and identification for those women whose ages greater than 35 years are crucial tasks to prevent pre-eclampsia

Types of occupation were found to be a risk factor for pre-eclampsia in this study. Housewife mothers are almost three times more likely to have pre-eclampsia (AOR = 2.76, 95%CI: 1.20, 4.32). This could be because the women who were always at home were prone to stress or psychological disturbance during early pregnancy, which leads to pre-eclampsia. Moreover, this could be due to a lack of knowledge about pregnancy and related issues such as pre-eclampsia.

Pregnancy alcohol consumption is another risk factor for pre-eclampsia. Mothers who consume alcohol while pregnant are more likely to develop pre-eclampsia (AOR = 1.53, 95% CI = 1.03, 2.04) than their counterparts. This could be due to uteroplacental malperfusion, which occurs during pregnancy and could also contribute to the link between alcohol consumption

and pre-eclampsia [48]. Therefore, mothers should also be advised to avoid drinking alcohol during pregnancy during their antenatal follow-up. However, the findings of this study contradict those of previous studies conducted in China, India, and the United Kingdom [49–51].

A history of pre-eclampsia is an important determinant of pre-eclampsia. In this meta-analysis, women with a history of pre-eclampsia were shown to develop pre-eclampsia, with the likelihood of its occurrence nearly twice as high as in women with no history of pre-eclampsia, and the pooled odd ratio showed that the association was statistically significant (AOR = 1.83, 95% CI = 1.24, 2.41). The findings are similar to those of a study conducted in Sub-Saharan Africa [52]. Similarly, having a family history of hypertension and a history of chronic hypertension may increase the risk of developing pre-eclampsia by nearly two and two-fold, respectively, when compared to women who have no such a history. Thus, according to the findings of this meta-analysis, a family history of hypertension and a history of chronic hypertension have a statistically significant association with preeclampsia. This report supports previous research conducted in Ethiopia, Sub-Saharan Africa, China, and the United States [40, 45, 52]. This could have happened because of genetic factors that contribute to the physiologic predisposition to pre-eclampsia.

Similarly, women with a history of diabetes mellitus were twice as likely as women without a family history of diabetes mellitus to develop pre-eclampsia (AOR = 2.02, 95%CI, 1.58, 2.47). Furthermore, women who have had multiple pregnancies can be twice as likely as their counterparts (AOR = 1.45, 95% CI = 1.09, 1.8) to develop pre-eclampsia. Thus, the results of this meta-analysis show that a family history of diabetes and a history of multiple pregnancies have a statistically significant association with pre-eclampsia. This report is comparable to studies conducted in Ethiopia, Sweden, China, the Republic of Korea, and Thailand [53–62]. The possible reason could be due to the fact that in cases of diabetes mellitus and multiple pregnancies cause an increase in placental mass or hypoxia of the placenta that possibly leads to the secretion of antiangiogenic factors like; tyrosine kinase 1 and endoglin sEng that antagonize placental growth factors and vascular endothelial growth factors result in hypertension, protein and maternal syndromes [2, 40]. Furthermore, hyperinsulinemia stimulates vascular smooth muscle cell proliferation, increases acute sympathetic nervous system activity, and modifies transmembrane cation transport, as well as renal sodium retention, release of the potent vasoconstrictor angiotensin II, and associated endothelial dysfunction. All these changes may increase blood pressure and, as a result, pre-eclampsia [2]. Therefore health care providers should encourage mothers to have antenatal follow-up early in the pregnancy and those with history of pre-eclampsia, history of GDM, family history of DM, multiple pregnancy and family history of hypertension should be educated about the risk of pre-eclampsia.

Gravidity, age < 24 years old, BMI ≥ 30 kg/m$^2$, history of gestational diabetes mellitus and family history of diabetes mellitus were not associated with this current systematic review and meta-analysis. This finding is inconsistent with other studies [40, 41, 45, 47, 52, 63–68]. The number of studies included in this meta-analysis is small. This may be the reason for the absence of an association with pre-eclampsia.

In general, in order to decrease the burdens associated with pre-eclampsia; during patient diagnosis and management, clinicians will conduct detailed patient evaluations to identify the risk factors of pre-eclampsia and to develop a better treatment protocol. Similarly, the aforementioned risk factors should be considered as health education messages by health care providers to teach mothers to practice preventive measures.

## Limitation of the study

All primary studies included in this review were limited to some areas, and other regions may be underrepresented. This review considered only articles published in the English language,

which may have resulted in the omission of studies that could have been published in other languages. The search strategy may miss unpublished articles; publication bias is likely high. In addition, high statistical heterogeneity was seen in this review because the inclusion of articles with different methods and different outcomes of interest for the assessment of risk factors of preeclampsia. The included studies were facility-based that may have a slightly elevated prevalence of pre-eclampsia compared to population-based study; hence, the findings may not represent true prevalence in the community.

## Conclusions

The overall pooled prevalence of pre-eclampsia was high compared to those reported previously. Early detection and treatment for pre-eclampsia are needed for pregnant mothers aged $\geq$ 35 years old. Further studies to assess the association between gravidity, age < 24 years old, BMI $\geq$ 30 kg/m$^2$, history of gestational diabetes mellitus, family history of diabetes mellitus, and pre-eclampsia are also needed. Government and other stakeholders should give due attention to early screening and treatment of preeclampsia.

## Supporting information

**S1 Checklist. PRISMA 2009 checklist.**
(DOC)

**S1 Table. A searching strategy for preeclampsia and its determinants in Ethiopia.**
(DOCX)

**S2 Table. Newcastle-Ottawa quality assessment scale for observational studies to assess preeclampsia and its determinants in Ethiopia.**
(DOCX)

## Acknowledgments

The authors acknowledge the sources of all primary studies.

## Author Contributions

**Conceptualization:** Bekalu Getnet Kassa.

**Data curation:** Bekalu Getnet Kassa, Sintayehu Asnkew, Azezu Asres Nigussie, Basaznew Chekol Demilew, Gedefaye Nibret Mihirete.

**Formal analysis:** Bekalu Getnet Kassa, Sintayehu Asnkew, Alemu Degu Ayele, Basaznew Chekol Demilew.

**Funding acquisition:** Bekalu Getnet Kassa, Azezu Asres Nigussie.

**Investigation:** Bekalu Getnet Kassa, Alemu Degu Ayele, Basaznew Chekol Demilew, Gedefaye Nibret Mihirete.

**Methodology:** Bekalu Getnet Kassa, Sintayehu Asnkew, Azezu Asres Nigussie, Basaznew Chekol Demilew.

**Project administration:** Bekalu Getnet Kassa, Alemu Degu Ayele, Gedefaye Nibret Mihirete.

**Resources:** Bekalu Getnet Kassa, Sintayehu Asnkew, Alemu Degu Ayele, Azezu Asres Nigussie, Basaznew Chekol Demilew, Gedefaye Nibret Mihirete.

**Software:** Bekalu Getnet Kassa, Basaznew Chekol Demilew, Gedefaye Nibret Mihirete.

**Supervision:** Bekalu Getnet Kassa, Sintayehu Asnkew, Alemu Degu Ayele, Azezu Asres Nigussie, Basaznew Chekol Demilew, Gedefaye Nibret Mihirete.

**Validation:** Bekalu Getnet Kassa, Alemu Degu Ayele, Basaznew Chekol Demilew.

**Visualization:** Bekalu Getnet Kassa, Sintayehu Asnkew, Alemu Degu Ayele, Azezu Asres Nigussie, Gedefaye Nibret Mihirete.

**Writing – original draft:** Bekalu Getnet Kassa, Azezu Asres Nigussie, Gedefaye Nibret Mihirete.

**Writing – review & editing:** Bekalu Getnet Kassa, Sintayehu Asnkew, Alemu Degu Ayele, Azezu Asres Nigussie, Basaznew Chekol Demilew, Gedefaye Nibret Mihirete.

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
