## [Decision Letter · Decision Letter 0]

10 Apr 2023

PONE-D-23-03153Preeclampsia and its determinants in Ethiopia: Systematic review and meta-analysisPLOS ONE

Dear Dr. Kassa,

Thank you for submitting your manuscript to PLOS ONE. After careful consideration, we feel that it has merit but does not fully meet PLOS ONE’s publication criteria as it currently stands. Therefore, we invite you to submit a revised version of the manuscript that addresses the points raised during the review process. While the efforts made by the authors in preparing this manuscript are to be appreciated, I would encourage them to address all the editor and reviewer comments to the best of their abilities. In addition, please make sure to address the query regarding the novelty and clinical usefulness of this article compared with the ones published before since this is a major issue. 

We look forward to receiving your revised manuscript.

Kind regards,

Aymen Ahmed

Guest Editor

PLOS ONE

Journal Requirements:

**Additional Editor Comments:**

Given that there are similar systematic reviews and meta-analyses conducted on this topic, can the authors please justify the need for this study to be conducted? What is the novelty of their research? How is it different from prior meta-analyses and what is it significantly adding to the current literature? Also please add the clinical implications of your findings.

Reviewers' comments:

Reviewer's Responses to Questions

**Comments to the Author**

1. Is the manuscript technically sound, and do the data support the conclusions?

Reviewer #1: Yes

Reviewer #2: Partly

2. Has the statistical analysis been performed appropriately and rigorously? 

Reviewer #1: Yes

Reviewer #2: Yes

3. Have the authors made all data underlying the findings in their manuscript fully available?

Reviewer #1: Yes

Reviewer #2: Yes

4. Is the manuscript presented in an intelligible fashion and written in standard English?

Reviewer #1: Yes

Reviewer #2: No

5. Review Comments to the Author

Reviewer #1: Authors conducted a meta-analysis to determine the prevalence and determinants of preeclampsia in Ethiopia. Manuscript is well written; tables and figures are well made. I have few comments to improve the current version.

Abstract

1. Please replace “therefore” or “hence” with “as a result” in the background section.

2. The following sentences can be merged for sake of conciseness and clarity.

“A systematic review and meta-analysis were conducted on observational facility-based studies between the years January 1, 2013, and January 1, 2023, in Ethiopia. Electronic searches such PubMed, Google Scholar, HINAR, Scopus, Web of Sciences and grey literature were used to 37 search primary studies.”

3. Use of the word “such as” gives an impression as if not all the databases that were searched to identify articles are mentioned. If that is the case, please mention all the databases in methods or eliminate the phrase.

4. Please mention the name of “Standardized measurement tool” that was used to extract data.

5. “STATA 11 statistical software was used to analyze the data as well and I2 tests evaluated study heterogeneity”

Please delete as well from this sentence.

6. Were there p-values along with AOR? They maybe worth mentioning in the results.

7. Conclusion is too big for abstract please try stating conclusions in maximum of 2-3 short sentences. This conclusion is too big for whole manuscript as well. Consider rephrasing.

Introduction

1. It might be worth mentioning exact figures of worldwide prevalence of preeclampsia.

2. Current version of introduction is very long and provides very general details regarding preeclampsia. Please shorten it and focus more towards building rationale of this particular study.

Methods

1. All studies were published in Ethiopia? Or all studies were conducted in Ethiopia?

2. Please state the criteria for including studies in a paragraph form.

3. Why were studies not taken from inception and taken from January 2013?

Results

1. Results are well described. Please mention the t value for egger’s test if it is available.

2. Replace Magnitude of Preeclampsia with Magnitude of Preeclampsia Prevalence or equivalent.

3. Semi colon is not correctly used. Please correct grammar throughout.

4. Please mention p-values alongside AOR for determinants, if available.

Conclusion

1. Please shorten the conclusion. Currently it provides a lot of redundant details. It should be one single paragraph, providing all the takeaway points.

Reviewer #2: 1. Please structure the Methods section under the headings of Search strategy, Study selection criteria, Screening process and data extraction, Extracted parameters, Quality assessment (ROB), and Statistical Analysis.

2. Please indicate how the screening of articles was done in more detail such as which reviewers performed the screening.

3. Kindly explain the rationale behind “Variables were chosen for this study if they were reported as a significant factor in two or more studies.”

4. In line 130, please make the phrase “up to December January 1, 2023” clearer.

5. The part concerning operational definitions should be mentioned under the heading of “Extracted parameters”.

6. There are grammatical errors in various places throughout the text, please correct them.

7. State the full forms of all abbreviations the first time they are used in the text.

8. Please be precise while wording the results section.

9. Kindly improve the language of the Results section, while correcting for any grammatical and spelling mistakes. For example, in Lines 216 - 222.

10. Edits to be made to Line 62 - pregnancy induced hypertension and Line 72 – multiparous.

11. Kindly shorten the Introduction by avoiding repetition and by making it more concise.

12. Please remove/replace all references older than 10 years.

6. PLOS authors have the option to publish the peer review history of their article (what does this mean?). If published, this will include your full peer review and any attached files.

Reviewer #1: No

Reviewer #2: No

---

## [Author Response · Author response to Decision Letter 0]

14 Apr 2023

A point-by-point response to the reviewers

Please ensure that your manuscript meets PLOS ONE's style requirements, including those for file naming. The PLOS ONE style templates can be found at https://journals.plos.org/plosone/s/file?id=wjVg/PLOSOne_formatting_sample_main_body.pdf and https://journals.plos.org/plosone/s/file?id=ba62/PLOSOne_formatting_sample_title_authors_affiliations.pdf

Response: we accept the comment and tried to correct based on the journal submission guideline.

Additional Editor Comments:

Given that there are similar systematic reviews and meta-analyses conducted on this topic, can the authors please justify the need for this study to be conducted? What is the novelty of their research? How it is different from prior meta-analyses and what is it significantly adding to the current literature? Also, please add the clinical implications of your findings.

Response: we accept your comment, but the previous systematic review and meta-analysis were conducted generally in the hypertensive disorder of pregnancy before three and five years respectively, while this study only focuses only on preeclampsia including the recently published articles in order to find the exact recent figures. It is the first study in Ethiopia and it shows preeclampsia is still prevalent. The clinical implication of this study is the prevalence of preeclampsia is still high; the clinician should be focused on prevention strategies for preeclampsia. In addition, during patient diagnosis and management, clinicians will conduct detailed patient evaluations to identify the risk factors of pre-eclampsia and to develop a better treatment protocol. For more information see the conclusion part.

Reviewer #1: Authors conducted a meta-analysis to determine the prevalence and determinants of preeclampsia in Ethiopia. Manuscript is well written; tables and figures are well made. I have few comments to improve the current version.

Abstract

1. Please replace “therefore” or “hence” with “as a result” in the background section.

Response: we accept your comment and updated the main document based on the given direction.

2. The following sentences can be merged for sake of conciseness and clarity.

“A systematic review and meta-analysis were conducted on observational facility-based studies between the years January 1, 2013, and January 1, 2023, in Ethiopia. Electronic searches such PubMed, Google Scholar, HINAR, Scopus, Web of Sciences and grey literature were used to 37 search primary studies.”

Response: we accept your comment and amend the document accordingly.

3. Use of the word “such as” gives an impression as if not all the databases that were searched to identify articles are mentioned. If that is the case, please mention all the databases in methods or eliminate the phrase.

Response: we accept your comment and amend the document accordingly.

4. Please mention the name of “Standardized measurement tool” that was used to extract data.

Response: we accept the comment, and this standardized measurement tool is a Microsoft Excel forms consisting of the author name, year of publication, study setting, sample size, study design, population, response rate, prevalence of preeclampsia and factors of preeclampsia.

5. “STATA 11 statistical software was used to analyze the data as well and I2 tests evaluated study heterogeneity”

Please delete as well from this sentence.

Response: we accept the comment and amend the document accordingly.

6. Were there p-values along with AOR? They maybe worth mentioning in the results.

Response: we accept the comment, but did not necessarily to put p-value once mention the adjusted odd ratio; which is more informative.

7. Conclusion is too big for abstract please try stating conclusions in maximum of 2-3 short sentences. This conclusion is too big for whole manuscript as well. Consider rephrasing.

Response: we accept the comment and updated based on your direction.

Introduction

1. It might be worth mentioning exact figures of worldwide prevalence of preeclampsia.

Response: we accept your comment and recommendation, and the prevalence of preeclampsia difference across each counties with different risk factors; we tried to put the figure in the main document.

2. Current version of introduction is very long and provides very general details regarding preeclampsia. Please shorten it and focus more towards building rationale of this particular study.

Response: we accept the comment and recommendation and amend the introduction accordingly.

Methods

1. All studies were published in Ethiopia? Or all studies were conducted in Ethiopia?

Response: all studies were conducted in Ethiopia.

2. Please state the criteria for including studies in a paragraph form.

Response: we accept the comment and recommendation and updated based on the given information.

3. Why were studies not taken from inception and taken from January 2013?

Response: 

- In recent; healthcare facilities, skilled birth attendants, and diagnostic modalities are better to identify preeclampsia. Due to this situation, we use the recent ten years of articles.

- Scientifically recommended that we use the recent ten-year published article to generalize the findings of the study, based on this we are starting from 2013. 

- As the healthcare infrastructure improved, service also improved leading to reporting such types of cases, and better representing the recent ten years than before.

Results

1. Results are well described. Please mention the t value for egger’s test if it is available.

Response: we accept the comment and recommendation, and t value for Egger’s test= 8.30, p > │t│=0.000

2. Replace Magnitude of Preeclampsia with Magnitude of Preeclampsia Prevalence or equivalent.

Response: We accept the comment and recommendation, and update based on the given direction

3. Semi-colon is not correctly used. Please correct grammar throughout.

Response: 

4. Please mention p-values alongside AOR for determinants, if available.

Response: We accept the comment and recommendation and tried to update the main document based on the given direction.

Conclusion

1. Please shorten the conclusion. Currently, it provides a lot of redundant details. It should be one single paragraph, providing all the takeaway points.

Response: we accept the comment and recommendation and corrected accordingly.

Reviewer #2: 1. Please structure the Methods section under the headings of Search strategy, Study selection criteria, Screening process and data extraction, Extracted parameters, Quality assessment (ROB), and Statistical Analysis.

Response: we accept your comment and recommendation and we tried to correct accordingly.

2. Please indicate how the screening of articles was done in more detail such as which reviewers performed the screening.

Response: Accept the comment and done based on your given direction.

3. Kindly explain the rationale behind “Variables were chosen for this study if they were reported as a significant factor in two or more studies.”

Response: we accept your comment and this sentence message is in order to compute the odd ratio of the pooled estimate the included studies at least have one common associated factor.

4. In line 130, please make the phrase “up to December January 1, 2023” clearer.

Response: we accept the comment and corrected it accordingly.

5. The part concerning operational definitions should be mentioned under the heading of “Extracted parameters”.

Response: we accept the comment and amend it accordingly.

6. There are grammatical errors in various places throughout the text, please correct them.

Response: accept and we tried to rephrase and rewrite by using different online software.

7. State the full forms of all abbreviations the first time they are used in the text.

Response: Accept the comment and done based on your given direction.

8. Please be precise while wording the results section.

Response: we accept your comment and recommendation and we tried to correct them accordingly.

9. Kindly improve the language of the Results section, while correcting for any grammatical and spelling mistakes. For example, in Lines 216 - 222.

Response: we accept the comment and amended it accordingly.

10. Edits to be made to Line 62 – pregnancy-induced hypertension and Line 72 – multiparous.

Response: We accept the comment and amended accordingly.

11. Kindly shorten the Introduction by avoiding repetition and by making it more concise.

Response: We reviewed based on your comment and corrected/ amended it accordingly.

12. Please remove/replace all references older than 10 years.

Response: we accept the comment and tried to amend the paper based on the given direction.

---

## [Decision Letter · Decision Letter 1]

3 May 2023

PONE-D-23-03153R1Preeclampsia and its determinants in Ethiopia: A systematic review and meta-analysisPLOS ONE

Dear Dr. Kassa,

Thank you for submitting your manuscript to PLOS ONE. After careful consideration, we feel that it has merit but does not fully meet PLOS ONE’s publication criteria as it currently stands. Therefore, we invite you to submit a revised version of the manuscript that addresses the points raised during the review process.

We look forward to receiving your revised manuscript.

Kind regards,

Aymen Ahmed

Guest Editor

PLOS ONE

Journal Requirements:

Additional Editor Comments:

The authors have responded to the query regarding clinical implications of this article. However, no significant text pertaining to implications has been added to the main text of the manuscript. Kindly add clinical implication details to the discussion section of the manuscript wherever the authors feel it is appropriate. Please ensure that the flow and coherence of the discussion text is maintained.

Reviewers' comments:

Reviewer's Responses to Questions

**Comments to the Author**

1. If the authors have adequately addressed your comments raised in a previous round of review and you feel that this manuscript is now acceptable for publication, you may indicate that here to bypass the “Comments to the Author” section, enter your conflict of interest statement in the “Confidential to Editor” section, and submit your "Accept" recommendation.

Reviewer #1: (No Response)

Reviewer #2: (No Response)

2. Is the manuscript technically sound, and do the data support the conclusions?

Reviewer #1: (No Response)

Reviewer #2: Yes

3. Has the statistical analysis been performed appropriately and rigorously? 

Reviewer #1: (No Response)

Reviewer #2: Yes

4. Have the authors made all data underlying the findings in their manuscript fully available?

Reviewer #1: (No Response)

Reviewer #2: (No Response)

5. Is the manuscript presented in an intelligible fashion and written in standard English?

Reviewer #1: Yes

Reviewer #2: (No Response)

6. Review Comments to the Author

**Reviewer #1:** All of my comments have been addressed except for few

1. Request to mention p-values alongside AOR throughout the manuscript. Authors mention that p-values are not required with AOR, which is not correct. P-value is absolutely needed to determine whether the stated association is statistically significant or just by chance and hence ignorable. Therefore, it is advised to please mention p-values throughout.

2. In abstract please explicitly mention you used Microsoft Excel. "Standardized data collection tool" is not sufficient information.

3. In methods section, please specifically mention the reason you included article between 2013 and 2023.

**Reviewer #2:** The authors have improved upon the quality of their article. I have some minor comments to make, after which I believe the article would merit publication. These comments are as follows;

1. I believe the Introduction section can still be restructured in a better manner.

2. Line 75-76: Rephrase as "Global studies reveal that the incidence of pre-eclampsia in nulliparous females ranged from 3 to 10% while in multiparous women, the incidence of preeclampsia ranged from 2 to 5%"

3. Line 91 - It should just be stated that "Preeclampsia and eclampsia are associated with hypertension and are known to poorly impact maternal and newborn mortality and morbidity"

4. In Lines 93-95, the first sentence and the phrase, "increases caesarian delivery rate", need to be rephrased.

5. Line 99 - The term "hypertension problems in women" needs to be better phrased.

6. Line 109 - Is it "across each region" or across various regions in the state?

7. Line 132 - Replace "whereas" with meanwhile.

8. Line 166 - Rephrase "through a two investigator (BCD and SA)"

9. Line 168 - kindly mention that it is an MS Excel sheet that includes (not "considers") data concerning the variables stated in the text.

10. Line 199 - Please replace "done to find" with "performed to ascertain"

11. Line 205 - Please edit it as "Of these articles, 972 duplicates were eliminated while 1536 articles were reviewed further for inclusion"

12. Line 216 - Specify Figure or Table along with S2. Additionally, exclude "see" from all brackets in the entire text where you are referring to a table or figure.

13. Line 220 - It should be "reported" instead of "report"

14. Line 225 - Use indicating instead of "showing".

15. Line 227 - Incorporate the t value for Egger's regression test.

7. PLOS authors have the option to publish the peer review history of their article (what does this mean?). If published, this will include your full peer review and any attached files.

Reviewer #1: **Yes: **Muhammad Sameer Arshad

Reviewer #2: **Yes: **Warda Ahmed

---

## [Author Response · Author response to Decision Letter 1]

5 May 2023

Point by point response to the editor and authors

Journal Requirements:

Response: we accept the comment and there is no used retracted article in the reference list.

Additional Editor Comments:

The authors have responded to the query regarding clinical implications of this article. However, no significant text pertaining to implications has been added to the main text of the manuscript. Kindly add clinical implication details to the discussion section of the manuscript wherever the authors feel it is appropriate. Please ensure that the flow and coherence of the discussion text is maintained.

Response: we accept the comment and we tried to update the main document accordingly.

Reviewer #1: All of my comments have been addressed except for few

1. Request to mention p-values alongside AOR throughout the manuscript. Authors mention that p-values are not required with AOR, which is not correct. P-value is absolutely needed to determine whether the stated association is statistically significant or just by chance and hence ignorable. Therefore, it is advised to please mention p-values throughout.

Response: we accept the comment, and update based on the given information.

2. In the abstract, please explicitly mention you used Microsoft Excel. "Standardized data collection tool" is not sufficient information.

Response: we accept the comment and amend based on the given direction.

3. In the methods section, please specifically mention the reason you included the article between 2013 and 2023.

Response: based on the following reason we restricted data extraction from 2013; 

- In recent; healthcare facilities, skilled birth attendants, and diagnostic modalities are better to identify preeclampsia. Due to this situation, we use the recent ten years of articles to generate the evidence about pre-eclampsia.

- Scientifically recommended that we use the recent ten-year published article to generalize the findings of the clinical the study, based on this we are starting from 2013. 

- As the healthcare infrastructure improved, service also improved leading to reporting such types of cases, and better representing the recent ten years than before.

Reviewer #2: The authors have improved upon the quality of their article. I have some minor comments to make, after which I believe the article would merit publication. These comments are as follows;

1. I believe the Introduction section can still be restructured in a better manner.

Response: we accept the comment and recommendation and amend the document accordingly

2. Line 75-76: Rephrase as "Global studies reveal that the incidence of pre-eclampsia in nulliparous females ranged from 3 to 10% while in multiparous women, the incidence of preeclampsia ranged from 2 to 5%"

Response: We accept the comment and corrected based on the given information.

3. Line 91 - It should just be stated "Preeclampsia and eclampsia are associated with hypertension and are known to poorly impact maternal and newborn mortality and morbidity"

Response: We accept the comment and amend based on the given information

4. In Lines 93-95, the first sentence and the phrase, "increases caesarian delivery rate", need to be rephrased.

Response: We accept the comment and tried to rephrase it accordingly

5. Line 99 - The term "hypertension problems in women" needs to be better phrased.

Response: we accept the comment and rephrased based on the given direction.

6. Line 109 - Is it "across each region" or across various regions in the state?

Response: amended as ‘across various regions in the state.’

7. Line 132 - Replace "whereas" with meanwhile.

Response: replaced with the recommended one

8. Line 166 - Rephrase "through a two investigator (BCD and SA)"

Response: we accept the comment and rephrased based on the given direction.

9. Line 168 - kindly mention that it is an MS Excel sheet that includes (not "considers") data concerning the variables stated in the text.

Response: Response: we accept the comment and update the document based on the given direction.

10. Line 199 - Please replace "done to find" with "performed to ascertain"

Response: corrected based on the given direction

11. Line 205 - Please edit it as "Of these articles, 972 duplicates were eliminated while 1536 articles were reviewed further for inclusion"

Response: We accept the comment and replaced it

12. Line 216 - Specify Figure or Table along with S2. Additionally, exclude "see" from all brackets in the entire text where you are referring to a table or figure.

Response: done it based on the given direction

13. Line 220 - It should be "reported" instead of "report"

Response: changed based on your recommendation

14. Line 225 - Use indicating instead of "showing"

Response: updated the document based on your recommendation.

15. Line 227 - Incorporate the t value for Egger's regression test.

Response: we accept the comment and add the t-value in the document based on the given direction.

---

## [Decision Letter · Decision Letter 2]

30 May 2023

Preeclampsia and its determinants in Ethiopia: A systematic review and meta-analysis

PONE-D-23-03153R2

Dear Dr. Kassa,

We’re pleased to inform you that your manuscript has been judged scientifically suitable for publication and will be formally accepted for publication once it meets all outstanding technical requirements.

Kind regards,

Aymen Ahmed

Guest Editor

PLOS ONE

Additional Editor Comments (optional):

Reviewers' comments:

Reviewer's Responses to Questions

**Comments to the Author**

1. If the authors have adequately addressed your comments raised in a previous round of review and you feel that this manuscript is now acceptable for publication, you may indicate that here to bypass the “Comments to the Author” section, enter your conflict of interest statement in the “Confidential to Editor” section, and submit your "Accept" recommendation.

Reviewer #1: All comments have been addressed

Reviewer #2:  All comments have been addressed

2. Is the manuscript technically sound, and do the data support the conclusions?

Reviewer #1: Yes

Reviewer #2: (No Response)

3. Has the statistical analysis been performed appropriately and rigorously? 

Reviewer #1: Yes

Reviewer #2: (No Response)

4. Have the authors made all data underlying the findings in their manuscript fully available?

Reviewer #1: (No Response)

Reviewer #2: (No Response)

5. Is the manuscript presented in an intelligible fashion and written in standard English?

Reviewer #1: Yes

Reviewer #2: (No Response)

6. Review Comments to the Author

Reviewer #1: (No Response)

Reviewer #2: The authors have addressed all my comments. 

7. PLOS authors have the option to publish the peer review history of their article (what does this mean?). If published, this will include your full peer review and any attached files.

Reviewer #1: No

Reviewer #2: **Yes: **Warda Ahmed

---

## [Editor Report · Acceptance letter]

2 Jun 2023

PONE-D-23-03153R2 

Preeclampsia and its determinants in Ethiopia: A systematic review and meta-analysis 

Dear Dr. Kassa:

I'm pleased to inform you that your manuscript has been deemed suitable for publication in PLOS ONE. Congratulations! Your manuscript is now with our production department. 

Kind regards, 

on behalf of

Dr. Aymen Ahmed 

Guest Editor

PLOS ONE